

# Abrupt cold events in the North Atlantic in a transient Holocene simulation

Andrea Klus[1], Matthias Prange[1], Vidya Varma[2], Louis Bruno Tremblay[3], Michael Schulz[1]

[1]MARUM - Center for Marine Environmental Sciences and Faculty of Geosciences, University of Bremen, Germany
[2]National Institute of Water and Atmospheric Research, Wellington, New Zealand
[3]Department of Atmospheric and Oceanic Sciences, McGill University, Montreal, Canada

*Correspondence to*: Andrea Klus (aklus@marum.de)

**Abstract.** Abrupt cold events have been detected in numerous North Atlantic climate records from the Holocene. Several mechanisms have been discussed as possible triggers for these climate shifts persisting decades to centuries. Here, we
describe two cold events that occurred during an orbitally forced transient Holocene simulation using the Community Climate System Model version 3. Both events occurred during the late Holocene (event 1 referring to 4305-4267 BP and event 2 referring to 3046-3018 BP) and were characterized by substantial surface cooling (-2.7 and -2.2 °C, respectively) and freshening (-0.7 and -0.6 PSU, respectively) as well as severe sea ice advance east of Newfoundland and south of Greenland. Sea ice even reached the Iceland Basin in the northeastern Atlantic at the climaxes of the cold events. Convection
and deep-water formation in the northwestern Atlantic collapsed during the events, while the Atlantic meridional overturning circulation was not significantly affected. The events were triggered by prolonged phases of a positive North Atlantic Oscillation which, through changes in surface winds, caused substantial changes in the sub-polar ocean circulation and associated freshwater transports, resulting in a weakening of the sub-polar gyre. Our results suggest a possible mechanism by which abrupt cold events in the North Atlantic region may be triggered by internal climate variability without the need of an
external (e.g. solar or volcanic) forcing.

## 1 Introduction

Holocene climate variability in the North Atlantic at different time scales has been discussed extensively during the past decades (e.g. Kleppin et al., 2015; Drijfhout et al., 2013; Hall et al., 2004; Schulz and Paul, 2002; Hall and Stouffer, 2001;
Bond et al., 1997, 2001; O'Brien et al., 1995; Wanner et al., 2001). North Atlantic cold events can be accompanied by sea ice drift from the Nordic Seas and the Labrador Sea towards the Iceland Basin as well as by changes in the Atlantic Meridional Overturning Circulation (AMOC). The sea ice proxy IP$_{25}$ (Belt et al., 2007) and diatom-based sea-surface temperature (SST) reconstructions from a sediment core north of Iceland show evidence for abrupt sea ice and climate changes (Massé et al., 2008). During the Little Ice Age (1300-1850 AD) several cold intervals at multi-decadal time scale
have been identified in Northern Hemispheric SST records (Crowley and Lowery, 2000), associated with the Dalton (1790-



1820 AD) and the Late Maunder solar minima (1675-1715 AD). Wanner et al. (1995) showed that the Late Maunder Minimum has been a relatively cool and dry period of approximately 40 years with a larger-than-normal sea ice extent. However not all North Atlantic cold phases during the Holocene are related to external forcing. For instance, Camenisch et al. (2016) report that the 1430s has been one of the coldest decades during the last millennium in north-western and central

Europe with a stronger-than-usual seasonal cycle in temperature neither related to anomalous solar nor volcanic activity.

Several mechanisms for the development of abrupt cold events in the North Atlantic have been discussed (e.g. Crowley, 2000; Alley, 2005). These include anomalous input of freshwater (Hawkins et al., 2011; Rahmstorf, 1996), volcanic activity (Sigl et al., 2015), solar forcing (Jiang et al., 2005; Steinhilber et al., 2009; Gray et al., 2010) or a combination of these factors (Büntgen et al., 2011; Jongma et al., 2007). Other causes for abrupt events that have been considered are associated

with internal atmosphere-ocean variability (Hall and Stouffer, 2001), sea ice transport (Wanner et al., 2008, and references therein) and sea ice-atmosphere interactions (Li et al., 2005, 2010). An expansion of sea ice could trigger a sudden and extensive change in air temperature by switching off the heat exchange between ocean and atmosphere (Kleppin et al., 2015). Furthermore, an abrupt climate shift can be forced by a 'Great Salinity Anomaly' (GSA) which originally describes an event with a major input of freshwater to the Nordic Seas in the 1960s and 1970s with a freshening of ~0.3 PSU and a

cooling of 1.5 °C off the central Greenland coast (Häkkinen, 1999) impacting the AMOC (Ionita et al., 2016). Hall and Stouffer (2001) described a similar event associated with an intensification of the East Greenland Current triggered by a high pressure anomaly centered over the Barents Sea that lasted ~40 years. Moreover, internal variability such as a positive phase of the North Atlantic Oscillation (NAO) can lead to a tripolar SST pattern in the North Atlantic (Deser et al., 2010; Hurrel et al., 2013). The surface above the sub-polar gyre (SPG) loses energy while the mid-latitudes gain energy as a result of the

associated wind anomalies. The duration of the positive NAO-phase is crucial for the response of the ocean (Lohmann et al., 2009; Visbeck et al., 2003). During a short positive NAO phase the SPG strengthens due to fast processes associated with surface fluxes, while it weakens if the phase continues over a longer period because of slow processes (e.g. transport of saline water or freshwater).

So far, our understanding of climate variability on multi-decadal to millennial timescales is too limited to explain if and to

what extent specific regions would be affected and whether rapid cold events are coupled to a substantial weakening of the AMOC. The understanding of multi-decadal cold events and their triggers will not only lead to a better understanding of paleoclimatic aspects but could potentially also improve predictions of climate change.

In the following, we study two spontaneous cold events in the northern North Atlantic and the Nordic Seas detected in a transient Holocene simulation with the Community Climate System Model version 3 (Varma et al., 2016). Focusing on

multi-decadal time scales we aim at improving our understanding of the mechanisms and feedbacks associated with Holocene cold events and their complex spatiotemporal pattern.



## 2 Model description and experimental design

A low-resolution transient simulation of Holocene climate change has been performed with the comprehensive Community Climate System Model version 3 (CCSM3) (Varma et al., 2016). Detailed information about the model including the source code is available at http://www.cesm.ucar.edu/models/ccsm3.0/. The fully coupled global climate model consists of four

components representing the atmosphere, ocean, land, and sea ice (Collins et al., 2006a). The atmospheric component of the CCSM3 is the Community Atmosphere Model version 3 (CAM3; Collins et al., 2006b). In this model set-up we used the T31 resolution (3.75° transform grid) with 26 unevenly distributed layers in the vertical (Yeager et al., 2006). The ocean component is the Parallel Ocean Program (POP; Smith and Gent, 2004) which has a nominal resolution of 3° with a refined meridional resolution of 0.9° around the equator and 25 vertical levels. CCSM3`s sea ice component is the Community Sea

Ice Model version 5 (CSIM5; Briegleb et al., 2004), which runs on the same horizontal grid as POP.

The model run has been performed at the North German Supercomputing Alliance (HLRN2) in Hanover. From a pre-industrial equilibrium simulation (Merkel et al., 2010) the model was integrated for 400 years with orbital forcing conditions representing 9000 years BP (before present) to reach a new quasi-equilibrium. Afterwards, a non-accelerated transient Holocene simulation was carried out by forcing the model with changing orbital parameters until the year 2000 BP (Varma

et al., 2016). Greenhouse gas concentrations, aerosol and ozone distributions were kept at pre-industrial values ($CH_4$ = 760 ppbv, $CO_2$ = 280 ppm, $N_2O$ = 270 ppbv; Braconnot et al., 2007). Variations in solar irradiance and changes in the continental ice-sheets are ignored. Variations in the orbital parameters were therefore the only external forcing in this model simulation. Surface freshwater fluxes (e.g. precipitation, evaporation, runoff, sea ice melting and formation) are represented through virtual salinity fluxes (Prange and Gerdes, 2006) with a reference salinity of 34.7 (equal to the global average salinity). The

model results presented below are all yearly mean values, unless otherwise specified. The definition of oceanic regions that we are referring to throughout the paper, and for which we have calculated average values is shown in Fig. 1.

## 3 Results

### 3.1 General state description

In the 7000-year-long Holocene model run two cold events are evident in the North Atlantic (Fig. 2a-e) for a time span of 39

25 years (4305-4267 BP) and 29 years (3046-3018 BP), respectively. We define cold event as a time span that exceeds the 2σ-interval of SST in the northwest Atlantic (NWA; Fig. 1). The time span 5000-4901 BP is exemplarily used to describe the "normal conditions". Both the SST and the sea-surface salinity (SSS) in the NWA show a negative trend over the Holocene with a mean of 4.3 °C and 34.2 PSU (Fig. 2a-b). Except for the abrupt cold events, the Iceland Basin is mostly ice-free (Fig. 2c). The NWA features a low annual mean sea ice concentration of 5.1 % with a slightly positive trend during the Holocene,

in line with the decreasing annual SSS and SST. Deep-water formation takes place in the NWA south of Greenland, the Irminger Sea and in the Nordic Seas (Fig. 1) around 75 °N | 5 °W. The averaged mixed-layer depth is 128 m in the NWA





and does not exhibit a trend over time (Fig. 2d). The areas of deep-water formation are ice-free while the Baffin Bay, the Labrador coastline and the northern Nordic Seas are partly ice-covered (not shown). The maximum AMOC streamfunction calculated north of 30°N and below 500 m water depth amounts to 13.8 Sv on average and varies between 9.4 and 17.7 Sv with a negative trend during the first ~1000 years (9k-8k BP) and no trend afterwards (Fig. 2e). Under normal conditions the

5 annual mean sea-level pressure difference between Iceland (1006 hPa) and the Azores (1022 hPa) amounts to 16 hPa (Fig. 3a) .

## 3.2 Anomalies during cold events

Two cold events are evident in the 7000-year-long time series: event 1 occurs between 4305 and 4267 BP (Fig. 4) and event 2 occurs between 3046 and 3018 BP (Fig. 5). During these cold events the NWA undergoes a substantial reorganization of

10 mixed-layer depth, sea ice, SST and SSS (Fig. 6a-h). In the following, all anomalies are described in comparison to the "normal conditions" of 5000-4901 BP. During events 1 and 2 the mixed-layer depths in the NWA, the Irminger Sea and the northern Nordic Seas decrease while the mixed-layer depth northeast of Iceland increases, indicating a southward shift of convection (Fig. 6a-b). The sea ice concentration in the Baffin Bay, the NWA, and the northern Nordic Seas increases whereas the sea ice concentration close to Iceland declines. In the NWA a maximum sea ice concentration of 37 % is

15 reached in 4294 BP and of 26 % in 3024 BP (Figs. 4a, 5a, 6c-d). This corresponds to anomalies of 26.1σ and 18.3σ during event 1 and 2, respectively for the NWA in comparison to the time span 5000-4901 BP. During some years, sea ice reaches the Iceland Basin with a maximum annual sea ice concentration of 4 % in year 4278 BP and 0.6 % in year 3021 BP. While the Atlantic does not show a significant change in SST or SSS south of 30 °N, the cold events result in a cooling of -2.7 °C during event 1 (-2.2 °C during event 2) and a freshening of -0.7 PSU during event 1 (-0.6 PSU during event 2) averaged over

20 the NWA (Figs. 2a-b, 6e-h). In the northern Nordic Seas, SST and SSS decrease, whereas the sea ice concentration increases (event 1: cooling of -2.1 °C, freshening of -1.1 PSU, rise in sea ice concentration of 34 %; event 2: cooling of -1.4 °C, freshening of -0.8 PSU, rise in sea ice concentration of 20 %; Fig. 5c-h). Northeast of Iceland, however, shows an increase in both SST and SSS, but a decline in sea ice concentration (event 1: warming of 1.6 °C, rise of 0.7 PSU, decline of -5 % in sea ice concentration; event 2: warming of 1.9 °C, rise of 0.9 PSU, decline of -11 % in sea ice concentration).

The AMOC time series shows a tendency to lower values during the first event (anomaly of -1.5 Sv), but still remains above 10 Sv (Figs. 2, 4). During the second event, the AMOC does not show a significant drop (Figs. 2, 5), although the mixed-layer depth in the NWA decreases during this period. Some solid and liquid freshwater transports in the polar and sub-polar region undergo a transition as well. In particular, the southward transport of liquid freshwater of the East Greenland Current through Denmark Strait intensifies by 495 km³/yr during event 1. During the second event the freshwater transport in the

NWA also increases but this time the pathway is different, i.e. the increase is observed through the Canadian Archipelago rather than through Denmark Strait. The freshwater transport in the East Greenland Current does not intensify much (68

km$^3$/yr) but the export of liquid freshwater through the Canadian Archipelago increases by 516 km$^3$/yr (Fig. 7b; event 1: 146 km$^3$/yr).

During event 1 (event 2) a weakening of the SLP above Greenland and Iceland of up to -1.1 hPa (-2.4 hPa) is present indicating a positive NAO-like pattern (Fig. 3b-c). The changes in SLP are even more apparent during wintertime (DJF).

During event 1 anomalies of +4.1 hPa above Europe and the North Atlantic between 35°N and 60°N and -7.2 hPa above Greenland and Iceland occur (+3.0 and -7.9 hPa during event 2; not shown). The NAO, defined as the leading mode (principal component) of variability in the SLP, is in a positive phase during winter (DJF) with SLP anomalies above Iceland (the Azores) exceeding the 2σ-interval during the two cold events.

### 3.3 Development prior to the events

The SLP pattern already changes prior to the events, which can be seen in the NAO time series (Fig. 8a, c; the leading principal component was calculated from the SLP in the Atlantic sector north of 35 °N). This behavior is most dominant in winter (DJF) for the first event and spring (MAM) for the second event. The NAO starts to increase towards a positive phase around 4320 BP (~15 years prior to event 1) for the first event and around 3070 BP (~24 years prior to event 2) for the second event while the mixed-layer depth in the NWA does not decrease until year 4302 and 3052 BP, respectively (Fig. 8b,

15   d). Most of the other ocean variables react more slowly with an almost linear trend (Figs. 4-5) and the following maximum anomaly.

Shortly prior to, during and after the events the SST and other variables show a quasi-oscillatory behavior, especially during event 1 (Figs. 4, 5). A frequency analysis showed that the period of the oscillations of the SST in the NWA is 20 years around event 1 and 14 years around event 2 (not shown).

### 4 Discussion

Given the previous evaluation that the change in the SLP associated with a positive trend in NAO happens prior to the change in the ocean variables suggests that the stochastic nature of the atmospheric variability is the trigger for both events. Along with the change of the SLP prior to the events the wind changes, e.g. the westerlies strengthen over the sub-polar North Atlantic, and southwestward anomalies of the wind between Iceland and Greenland (event 1) and northeastward

anomalies between Scotland and Iceland occur. This leads to a change in ocean circulation and freshwater fluxes as well as in sea ice concentration and SST and SSS patterns (Figs. 6-7, 9).



### 4.1 SST anomaly pattern and changes in ocean circulation

An observed tripolar SST pattern in the North Atlantic can result from wind fields associated with a positive phase of the NAO (Deser et al., 2010; Hurrell et al., 2013). Our SLP anomaly (Fig. 3b-c) as well as SST anomaly patterns (Fig. 6e-f) show similarities with this, while the range of anomaly is much broader in our case (4 °C SST anomaly amplitude in our
study compared to 0.5 °C in Deser et al., 2010). Deser et al. (2010) concluded that the NAO-related SST anomaly pattern is mainly driven by turbulent energy flux anomalies in the North Atlantic. The ocean loses energy above the sub-polar gyre (SPG) due to strengthened westerlies and gains energy in the mid-latitudes due to weakened wind. Lohmann et al. (2009) show a strengthened and cooled SPG due to a short positive NAO phase, but after ~10 years it becomes warmer and weaker, and it becomes weaker due to a negative NAO phase. They conclude a highly non-linear response of the circulation in the
North Atlantic to atmospheric forcing. In our study, the westerlies strengthen as well and the ocean loses heat above the SPG. A decreased density can be seen in the center of the SPG due to a cooling and freshening (Fig. 6e-h) as well as a weakening of its strength (Fig. 9) which shows similarities to the feedback loop about bistable behavior of the SPG suggested by Levermann and Born (2007). The advection of salty an warm water from the south into the SPG is reduced resulting in higher SSS at ~40 °N and lower SSS in the SPG (Fig. 9). A weakened SPG (Fig. 9) favors the cold and fresh
water to stay in the SPG and the warm and salty water in the NS and south of the SPG. As a consequence the density at the surface is decreasing and the density profile of the water column has a higher stability. The water in the NWA becomes so fresh that deep-convection is disturbed (Figs. 2, 4-6). In contrast to a temperature anomaly in the NWA of -1.5°C as reported in Lohmann et al. (2009), we found a SST anomaly of up to -4 °C, resulting in SST in the area of deep-water formation close to the freezing point. Therefore sea ice is less likely to melt and the high sea ice concentration hampers the interaction
between ocean and atmosphere further reducing the surface heat loss of the ocean.

The transport in the Nordic Seas is stronger towards ~75 °N form north and south. The southern Nordic Seas experience a cyclonic anomaly in ocean circulation leading to a saltier sea surface, while the northern Nordic Seas experience an anti-cyclonic anomaly associated with a freshening (Fig. 9). During event 2 the changes in ocean circulation and wind increase the transport of sea ice from the Canadian Shelf and Baffin Bay to the area of deep convection in the NWA. Other studies
revealed that a SST anomaly pattern similar as in our study can force a positive NAO phase (Czaja and Frankignoul, 2002; Frankignoul and Gastineau, 2015; Gastineau and Frankignoul, 2014). This would produce a positive feedback, where the positive NAO triggers such a SST anomaly pattern and vice versa, leading the cold climate state to maintain for decades.

### 4.2 Increase of freshwater transports

The mean liquid freshwater transports (reference salinity of 34.7 PSU) through Denmark Strait and the Canadian
Archipelago amount to -4077 km$^3$/yr and -2575 km$^3$/yr (negative values refer to southward transports), which fits into the range of published values, i.e. -4762 to -2712 km$^3$/yr and -3200 to -920 km$^3$/yr, respectively (Aagaard and Carmack, 1989;





Dickson et al., 2007; Jahn et al., 2010; Karcher et al., 2005; Oka et al., 2006; Prange and Gerdes, 2006). Prior to and during the events the freshwater fluxes through Denmark Strait and the Canadian Archipelago are intensified by changes in the surface wind. Southwestward wind anomalies along the East Greenland coast push more solid (Kwok, 2000) and liquid freshwater through Denmark Strait (event 1: liquid freshwater anomaly -495 km$^3$/yr after ~4320 BP; Fig. 7a). These findings

have similarities to the climate anomaly that lasted ~40 years in the sub-polar gyre, discussed in Hall and Stouffer (2001). They found a strong intensification of the East Greenland Current related to southwestward wind anomalies at the east coast of Greenland due to a high pressure anomaly over Greenland and the Barents Sea. The SLP and wind anomaly in their study show some differences to ours. In their study, the wind anomaly along the Greenland coast plays the most important role, while the strengthening of freshwater transport through Denmark Strait in our study indicates that the intensified and

northward shifted Iceland low is more relevant for triggering the cold event. While Hall and Stouffer (2001) found the largest SSS anomaly of ~ -1 PSU close to the east and southeast coast of Greenland, in our study the largest negative anomaly of up to -1.7 PSU can be seen in the northern Nordic Seas and the North Atlantic at around 48 °N (during event 2 the largest negative (positive) anomaly is about 1.1 PSU (1.4 PSU)). This is a lot more than during the Great Salinity Anomaly of the 1960s and 1970s with a freshening of about 0.2 to 0.3 PSU and a cooling of 1 to 1.5 °C off the central

Greenland coast. The salinity in the southern Nordic Seas and the North Atlantic at about 40 °N rises up to 1.6 PSU during the first event. A major reason for the difference in the patterns is the change in ocean circulation found in our results. Combining the change in ocean circulation along with the SST, SSS and sea ice anomalies (Fig. 6e-h) it can be stated that a stronger exchange between the North Atlantic and the Nordic Seas took place transporting more fresh and cold water out of Denmark Strait, especially during the first event. In contrast to the first event the anomalous isobars in the Baffin Bay are

aligned in the northwest-to-southeast direction (Fig. 3), leading to a southeastward wind anomaly. This way the Canadian Archipelago carries more freshwater (Fig. 7b) from the Arctic towards the NWA while the Denmark Strait transport decreases. The resulting SSS pattern is very similar to the first event and just differs in magnitude (Fig. 6g-h). The increased transports through Denmark Strait and the Canadian Archipelago freshen and cool the NWA and help to trigger and maintain the cold state with a fresh and cold NWA without deep-water formation.

## 4.3 Quasi-decadal oscillations

While the quasi-decadal oscillations, which are most dominant during event 1, are visible in all mentioned ocean variables (Figs. 4-5), the atmosphere does not show such an oscillatory behavior, as it can be seen in the time series of the principal components (Fig. 8a, c). This behavior is an indicator for a nearby tipping point of the ocean which characterizes the transition from a stable to an oscillatory mode (Scheffer et al., 2009). The weather as a fast component of the climate system

can be interpreted as a random forcing for slower components like the ocean (Hasselmann, 1976; Ashwin et al., 2012). Hence, relatively short atmospheric anomalies can serve as driving mechanism for climate fluctuations on long time scales ('*noise-induced tipping*', Ashwin et al., 2012).



### 4.4 AMOC and reduced deep-water formation

The AMOC time series does not undergo a climate transition to a weak state during the events (Figs. 4-5), although the mixed-layer depth in the NWA decreases during these periods. The increasing mixed-layer depths northeast of Iceland during the events seem to compensate for the decreased mixed-layer depths in the northern Nordic Seas and the NWA (Fig.

6a-b) keeping the AMOC in a strong state. Deep convection, however, does not occur in the Labrador Sea where it would be expected (Kuhlbrodt et al., 2007), but is shifted towards the east, south of Greenland. In accord with other simulations using the CCSM (Prange, 2008; Gnanadesikan et al., 2006) a too low surface salinity in the Labrador Sea could explain this spatial shift.

In disagreement with other modeling studies in which a positive NAO leads to a cooling and density increase of the upper

ocean layer in the northern North Atlantic (Delworth and Dixon, 2000; Häkkinen, 1999), the positive NAO phase during our events is neither accompanied by an intensification of deep convection in the Labrador Sea/NWA nor the AMOC. In our study, the freshening of the NWA leads the density to decrease and the extreme cooling of -4 °C results in an SST close to the freezing point. Consequently, the fresh and ice covered water column becomes stable and deep-water formation breaks down (Figs. 2, 4-5). A short phase of a positive NAO leads to changes in turbulent surface fluxes (fast processes) while a

persistent positive NAO phase over decades, as in our study, can change the ocean freshwater transports (slow processes). The forcing is the same (positive NAO) but the consequences depend on its duration. Furthermore, a severe weakening of deep-water formation such as in our study is often accompanied by a mode switch of the AMOC (Drijfhout et al., 2013; Schulz et al., 2007). In our study, the area of deep-water formation in the NWA cools substantially, while the AMOC weakens by only 1.5 Sv (0.6 Sv) during event 1 (event 2). These results are in accord with results from Born and Levermann

(2010) who show an abrupt transition in the North Atlantic due to a freshwater input of $160 \cdot 10^{12}$ m$^3$ within two years to the coast of the Labrador Sea, while the AMOC just weakens by 1.5 Sv and recovers after approximately 100 years.

### 4.5 Link to drift-ice events

The stronger and northward-shifted westerlies due to a positive NAO can transport sea ice from the NWA farther across the northern North Atlantic until almost west of Ireland (4295 BP; Figs. 4a, 5a). A comparable sea ice extent has been discussed

by Bond et al. (2001) based on ice-rafted debris. The authors stated that the sea ice in their study was probably transported by surface waters from north of Iceland towards the coring side of core MC-VM-29-191 west of Ireland. An atmospheric change with SLP anomaly representing a positive NAO-like state as in our study could have been the trigger for the events discussed in Bond et al. (2001). The authors also suggest that a decrease in SST and SSS during drift-ice events potentially led to a reduced deep-water formation in the North Atlantic and therefore a reduced thermohaline circulation. We suspect

that an increase in sea ice concentration in the southern Nordic Seas, unlike in our study, could possibly weaken the deep-





water formation there as well, such that its compensating effect on the reduced NWA deep-water formation would vanish, leading the AMOC to collapse as suggested by Bond et al. (2001).

## 5 Summary and conclusions

Two abrupt cold events that last for 39 and 29 years, respectively, have been detected in the northern North Atlantic during a Holocene model run. The events were initiated by a positive NAO phase and the associated wind anomalies, inducing chances in ocean circulation and freshwater transports in the sub-polar seas. The freshwater transport through Denmark Strait respectively Canadian Archipelago intensified leading the convection area in the NWA to become fresher and less dense at the surface and the water column to become more stable. The events are characterized by a cooling, freshening and

weakening of the SPG and a severe sea ice advance maintaining a circulation state without deep convection in the NWA. We suggest that the ocean-atmosphere coupling (SST anomaly tripole forces a positive NAO-like state and vice versa) and the increased freshwater transports through Denmark Strait and the Canadian Archipelago helped to maintain the cold state for decades. A return to normal NAO atmospheric conditions heralds the termination of the cold events.

Atmospheric variability together with a widely expanding sea ice concentration play a key role in the development of the

cold events and the associated changes in ocean circulation. It seems that the important key component in the complex spatiotemporal pattern of Holocene cold events is the sensitivity of the northern North Atlantic to the atmospheric and sea ice background state. This suggests that atmospheric anomalies (like a persistent positive NAO phase) may have led to climate transitions in the past and that similar processes may also be a major trigger for future climate variability. An important component that needs to be considered in this context is the extent of the sea ice. Due to climate warming, the sea

ice concentration in the Arctic realm massively decreases, leading to an increase of freshwater content in the northern North Atlantic due to melting sea ice, which would render the ocean more sensitive to an atmospheric trigger stopping the deep-water formation. On the other hand, sea ice is less likely to reach areas of deep-water formation in a warmer climate. As a result, the potential of a stochastic atmospheric anomaly to trigger an abrupt cold event may be modified by the changing North Atlantic background climate under increasing greenhouse gas emissions, which may further complicate the

investigation and projection of abrupt climate change.

## Data availability

Model results with CCSM3 presented in this study will be uploaded to the PANGAEA database.





**Competing interest**

The authors declare that they have no conflict of interest.

**Acknowledgements**

This project was supported by the Deutsche Forschungsgemeinschaft (DFG) through the International Research Training
Group "Processes and impacts of climate change in the North Atlantic Ocean and the Canadian Arctic" (IRTG 1904
ArcTrain). The authors would like to thank Ute Merkel for making available the spun-up pre-industrial control run restart
files. The CCSM3 experiments were performed with resources provided by the North-German Supercomputing Alliance
(HLRN).

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





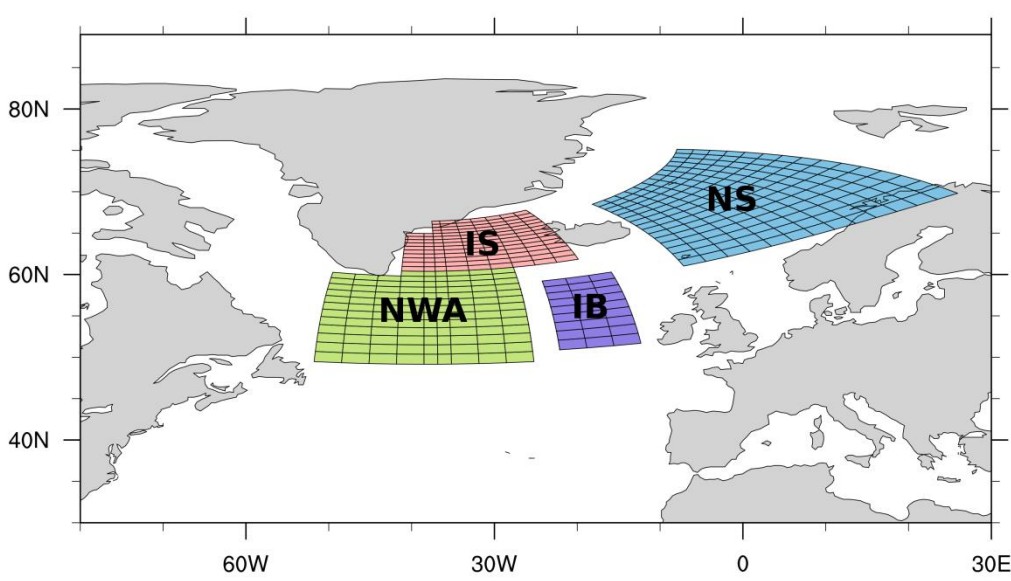

**Figure 1: Map showing areas of investigation: northwest Atlantic (NWA) in green, Irminger Sea (IS) in coral, Iceland Basin (IB) in purple and Nordic Seas (NS) in blue.**





Figure 2: Time series of the annual mean a) sea-surface temperature (SST; in °C) in the NWA, b) sea-surface salinity (SSS; in PSU) in the NWA, c) sea ice concentration in % in the Iceland Basin (green) and in the North West Atlantic (NWA; blue), d) mixed-layer depth (MLD; in m) in the NWA, e) AMOC calculated as the maximum of the streamfunction in the Atlantic north of 30°N and below 500 m water depth. The respective 40-year-running means are indicated by the light blue lines. The dashed grey line indicates the 2-standard-deviation range for the SST. The red lines indicate the time spans for the reference time, event 1 and event 2.







**Figure 3: a) Sea-level pressure during normal conditions (5000-4901 BP), sea-level pressure anomaly during b) event 1 (4305-4267 BP), and c) event 2 (3046-3018 BP).**





Figure 4: Zoom-in on event 1. Same fields as in Figure 2. The red box indicates the time span of the first event.





**Figure 5: Zoom-in on event 2. Same fields as in Figure 2. The red box indicates the time span of the second event.**





Figure 6: Anomaly maps for event 1 (left; 4305-4267 BP) and event 2 (right; 3046-3018 BP) of mixed-layer depth (m; a-b), sea ice concentration (%; c-d), sea surface temperature (°C; e-f) and sea surface salinity (PSU; g-h).



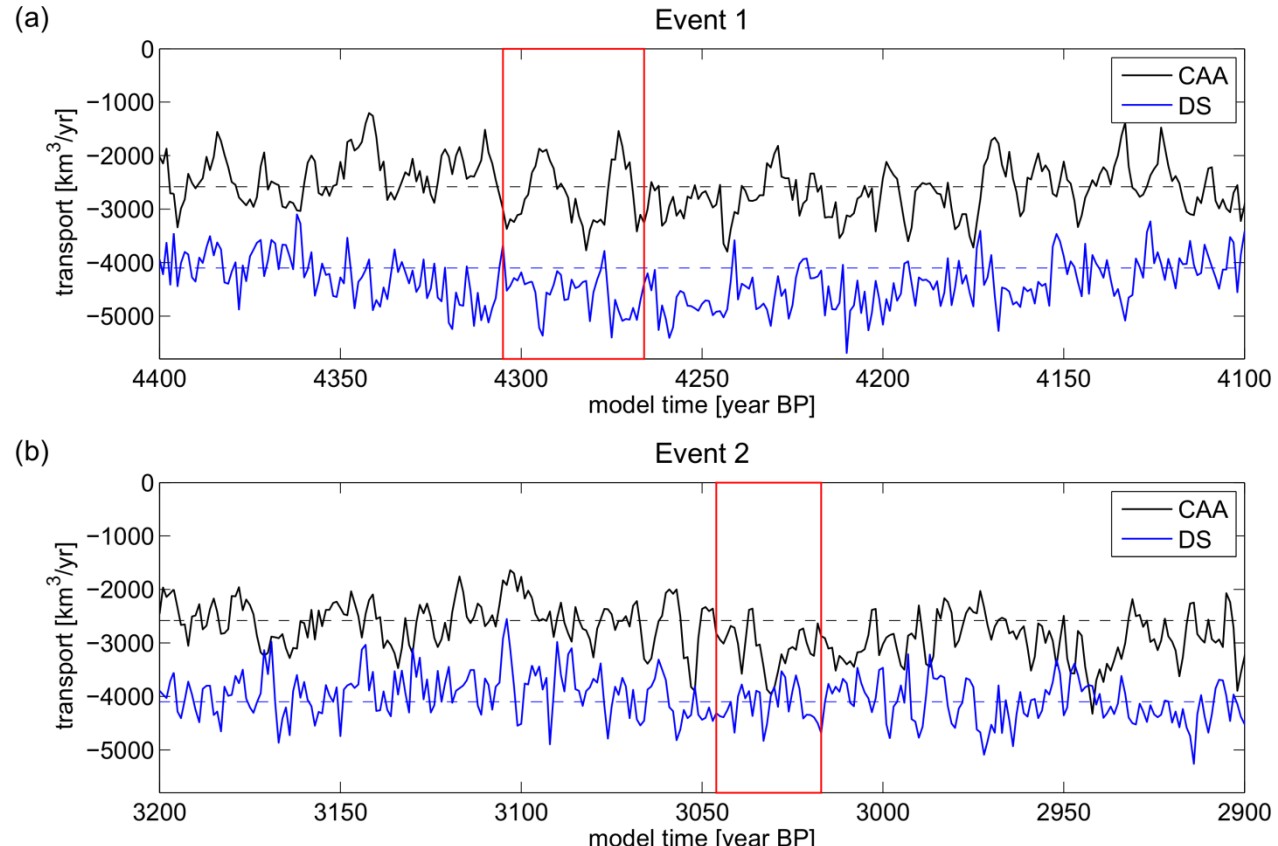

Figure 7: Freshwater flux through the Canadian Arctic Archipelago (CAA; black) and the Denmark Strait (DS; blue) with a reference salinity of 34.7 PSU during a) event 1 and b) event 2. Negative values correspond to a southward transport. The red boxes indicate the time span of the events.



**Figure 8: Leading principal component (PC) of the sea-level pressure over the Atlantic sector north of 35°N for a) event 1 during winter (DJF) and c) event 2 during spring (MAM). The respective 10-year-running means are indicated by the light blue lines. Annual mixed-layer depth for event 1 (b) and event 2 (d). The red boxes indicate the time span of the events.**



**Figure 9: a) Sea surface salinity (SSS) and ocean circulation for the top 100 m during normal conditions, and SSS anomaly and ocean circulation anomaly for top 100 m during b) event 1 and c) event 2.**