# Peer review of "Abrupt cold events in the North Atlantic in a transient Holocene simulation"

_Climate of the Past, 2017_

## Referee Comment (RC1) · C. C. Raible (Referee) · 10 Nov 2017

Summary

The authors analyze two abrupt cold events in the North Atlantic Ocean (sea surface temperature) simulated in a transient orbital-forced simulation of the Holocene. Given the experimental design the cold events are generated by internal variability rather than external forcing. The triggering mechanism is the North Atlantic Oscillation which shows a prolonged positive phase starting earlier than the ocean response, mainly via changes in the momentum transfer to the ocean and a subsequent weakening if the sub polar gyre circulation. Interestingly, the Atlantic Meridional Overturning circulation is not involved in the processes leading to these cold events.

[Figure]

The manuscript is nicely written and well structured. More importantly it discusses an important scientific question, namely the role of internal variability in coupled climate system generating extreme events in the ocean. The authors suggest an interesting mechanism as the atmosphere seems to play the triggering process for a long lasting sea surface temperature anomaly. Thus this study is certainly important for the interpretation of past climate states and proxy records, so I recommend possible publication of this study in Climate of the Past after minor revisions (see below).

Still one mayor comment needs to be dealt with: The authors have a very long simulation so they can check how often a similar NAO period has occurred which directly can answer the question whether the NAO is really the trigger or not. Assume the authors find 10 other periods of prolonged positive phase of the NAO but no cold event, then it is questionable whether the NAO is the only trigger necessary for such an event.

Minor Comments

Title: I suggest to include ocean after Atlantic to make clear that the cold events discussed are mainly found in the sea surface temperature. p1,l16: Statistically significant?

Introduction: As the NAO plays an important role in the mechanism proposed the authors need to give a brief review of current NAO reconstructions and their problems, e.g., Ortega et al. (2015, A multi-proxy model-tested NAO reconstruction for the last millennium. Nature, 523 71-75),

P2,l11: I think that the authors need to mention the study of Lehner et al. (2013, Amplified inception of European Little Ice Age by sea ice-ocean-atmosphere feedbacks. J. Climate, 26, 7586-7602) as they show another important possible mechanism explaining cold events in the North Atlantic.

P3,l20-21: Please change to "The definition of oceanic regions used throughout the paper is shown in Fig. 1."

P3,l25: "We define cold events as . . ."

P4,l22-24: Please change to "However, an increase in both SST and SSS is found northeast of Iceland, but a decline in sea ice concentration is detected there (event 1: warming of 1.6 °C, rise of 0.7 PSU, decline of -5 % in sea ice concentration; event 2: warming of 1.9 °C, rise of 0.9 PSU, decline of -11 % in sea ice concentration)."

P5,l8: I suggest to refer to Figure 8 here and add the 2 sigma range in Fig 8.

P6,l9: Please change to "They concluded . . ."

P6,l13: "The advection of salty and warm . . ."

P6,l21: "The transport from north to south is stronger towards . . . " Otherwise I do not understand the sentence.

P6,l26: please add e.g. before the references as there are many more publications which highlight the tripole structure and dealing with the feedback process of SSTs and the atmospheric circulation.

P7,l13: "This is substantially more than . . ." P7,l16: please include a line break. Section 4.2: Given your mechanism I think it is necessary to discuss the differences to the mechanism described by Lehner et al 2013 as these authors suggest a mechanism which does not rely on the NAO. This is even more important as these authors use the same model though in different resolution.

Section 4.3: These oscillations are interesting but how model dependent are they? Is this behavior found also in other simulations with the same model? Is the reason mainly the coarse resolution and or the fact that the AMOC is rather weak and close to a threshold?

P8,l20: "who showed "

Section 5: Given the fact that the model simulates to much sea ice how does this affect your proposed mechanism? Or in other words, is this mechanism only possible

because of the rather strong biases in sea ice distribution?

Fig. 1: Please start the caption with "Areas of "

Fig.2: The reference period seems to be selected in a rather warm period and roughly 1000 years after the second extreme event – why? I suggest to use a longer ref period just between the two extreme events so from 4000-3200. Or just two ref periods before or after the extreme event to avoid any orbital forcing signal.

Fig.3: It remains unclear how the anomalies are calculated. I guess it is the difference between mean over the period of an event and reference. In this case I would call it a difference4 and not an anomaly (which is normally a deviation to a long-term mean, rather than a difference between independent periods). Please change this throughout the text

Fig. 6 and 9: see Fig.3

---

## Short Comment (SC1) · 15 Nov 2017

This is indeed an interesting paper. I don't aim at giving an exhaustive review here, but I would suggest that the authors compare and discuss their results with those in Moreno-Chamarro et al. [2015].

My impression is that many of the features and mechanisms of the cold events described here with the CCSM3 model are actually very similar to those of the decadal cold events in the simulations with the MPI-ESM model in that referred paper. For example, the conspicuous cooling and sea ice expansion around the Labrador Sea, the weakening of the subpolar gyre and the oceanic deep convection shutdown, or the length of the cold events itself. I would add, nonetheless, that the authors are here able

to go a step farther and identify a potential trigger mechanism of such events. Such discussion would be a very valuable contribution to the Klus et al.'s manuscript.

References

Moreno-Chamarro, E., Zanchettin, D., Lohmann, K., Jungclaus, J. H. (2015). Internally generated decadal cold events in the northern North Atlantic and their possible implications for the demise of the Norse settlements in Greenland. Geophysical Research Letters, 42(3), 908-915.

---

## Author Comment (AC1) · 30 Jun 2018

**Response to Referee #1 C.C. Raible:**

**Ref.: Ms. No. cp-2017-106**

**Title: Abrupt cold events in the North Atlantic in a transient Holocene simulation"**

We highly appreciate your time and effort giving constructive comments and suggestions, which will help us to greatly improve our manuscript. We have prepared a new version of the manuscript with your comments taken into account. Below we include a point-by-point reply to each comment with the original comments in black and our responses in blue.

**Referee #1 comments:**

The authors analyze two abrupt cold events in the North Atlantic Ocean (sea surface temperature) simulated in a transient orbital-forced simulation of the Holocene. Given the experimental design the cold events are generated by internal variability rather than external forcing. The triggering mechanism is the North Atlantic Oscillation which shows a prolonged positive phase starting earlier than the ocean response, mainly via changes in the momentum transfer to the ocean and a subsequent weakening if the sub polar gyre circulation. Interestingly, the Atlantic Meridional Overturning circulation is not involved in the processes leading to these cold events.

The manuscript is nicely written and well structured. More importantly it discusses an important scientific question, namely the role of internal variability in coupled climate system generating extreme events in the ocean. The authors suggest an interesting mechanism as the atmosphere seems to play the triggering process for a long lasting sea surface temperature anomaly. Thus this study is certainly important for the interpretation of past climate states and proxy records, so I recommend possible publication of this study in Climate of the Past after minor revisions (see below).

Still one mayor comment needs to be dealt with: The authors have a very long simulation so they can check how often a similar NAO period has occurred which directly can answer the question whether the NAO is really the trigger or not. Assume the authors find 10 other periods of prolonged positive phase of the NAO but no cold event, then it is questionable whether the NAO is the only trigger necessary for such an event.

We checked the entire time series of the leading principal component of the sea-level pressure above the North Atlantic region representing the NAO (winter and spring values; as in Fig. 8) for further events. The NAO during spring time (MAM) only exceeds the 2σ-interval for more than a 10 years during the second cold event. Except for the two cold events described in the manuscript the NAO during winter time (DJF) exceeds the 2σ-interval only once for more than 10 years during 7861-7874 BP (14 years) which is still shorter than during the two cold events (29 and 39 years) discussed in the paper. The maximum SST (SSS) anomaly during this event is -0.7 °C (-0.13 PSU; Fig. R1). Although the anomalies during this unreported event are smaller and the sea ice cover did not expand over the

NWA, the spatial patterns of SST, SSS, mixed-layer depth, and sea-ice cover anomalies show large similarities to the two cold events discussed in our paper (Fig. R2). Furthermore, the freshwater transport anomalies through Denmark Strait and the Canadian Arctic Archipelago reached -672 and +328 km$^3$/yr for a net transport anomaly of only -343 km$^3$/yr (not shown) compared to the anomalous net transports of about -700 km$^3$/yr for the two cold events.

During 7861-7874 BP the positive NAO phase was too short to lead to a sufficient freshening of the deep convection area in the NWA and therefore did not trigger an abrupt cold event. In conclusion, a SSS anomaly of ~ 0.5 PSU during the two reported cold events was enough to decrease deep convection in the NWA and lead to an abrupt cold event, while the SSS anomaly during 7861-7874 BP was insufficient to affect deep convection significantly. Considering the amount of SSS anomaly required to stop the deep convection (~ 0.5 PSU), the size of the NWA area, and its mean mixed-layer depth, we can roughly estimate the associated salt deficit to be in the order of 150 Gt. For reference, Dickson et al. (1988) estimated a salt anomaly of about 78 Gt for the Great Salinity Anomaly. We suggest that there exists a threshold in SSS that can be exceeded after ~ 20 years of anomalous atmospheric forcing and freshwater transports in the North Atlantic as well as through Denmark Strait and the Canadian Arctic Archipelago, and which affects deep convection in the North Atlantic significantly.

The information about the unreported NAO event has been added to the results section 3.2 and the discussion about the NAO and corresponding anomalies in the NWA and the freshwater transport anomalies have been added to the manuscript in section 4.2. We decided to not show the entire NAO and freshwater transport time series in the manuscript. Still the data will be uploaded to the PANGAEA database.

**Minor Comments**

Title: I suggest to include ocean after Atlantic to make clear that the cold events discussed are mainly found in the sea surface temperature. p1,l16: Statistically significant?

Title: done

p1,l16: yes, but we decided to use the word "substantially" here and to provide some numbers.

Introduction: As the NAO plays an important role in the mechanism proposed the authors need to give a brief review of current NAO reconstructions and their problems, e.g., Ortega et al. (2015, A multi-proxy model-tested NAO reconstruction for the last millennium. Nature, 523 71-75),

We agree that a short overview on NAO reconstructions is helpful in this context and added a brief summary in the introduction based on Ortega et al. (2015), Raible et al. (2014), Lehner et al. (2012), and Schmutz et al. (2000).

P2,l11: I think that the authors need to mention the study of Lehner et al. (2013, Amplified inception of European Little Ice Age by sea ice-ocean-atmosphere feedbacks. J. Climate, 26, 7586-7602) as they show another important possible mechanism explaining cold events in the North Atlantic.

done (here and in the discussion section)

P3,l20-21: Please change to "The definition of oceanic regions used throughout the paper is shown in Fig. 1."

done

P3,l25: "We define cold events as..."

done

P4,l22-24: Please change to "However, an increase in both SST and SSS is found northeast of Iceland, but a decline in sea ice concentration is detected there (event 1: warming of 1.6 _C, rise of 0.7 PSU, decline of -5 % in sea ice concentration; event 2: warming of 1.9 _C, rise of 0.9 PSU, decline of -11 % in sea ice concentration)."

done

P5,l8: I suggest to refer to Figure 8 here and add the 2 sigma range in Fig 8.

done. The caption of Fig. 8 has been changed accordingly.

P6,l9: Please change to "They concluded..."

done

P6,l13: "The advection of salty and warm..."

done

P6,l21: "The transport from north to south is stronger towards... " Otherwise I do not understand the sentence.

Thank you for pointing this out. We mean that in the Nordic Seas the transport towards ~ 75 °N comes from the north as well as from the south in the top 100 m. At ~75 °N deep-convection takes place. The sentence has been rewritten in section 4.1 of the revised manuscript. The new text is: " In the Nordic Seas the transport in the top 100 m from north and south towards ~75 °N (where deep-convection takes place) becomes stronger."

P6,l26: please add e.g. before the references as there are many more publications which highlight the tripole structure and dealing with the feedback process of SSTs and the atmospheric circulation.

done

P7,l13: "This is substantially more than..."

done

P7,l16: please include a line break.

done

Section 4.2: Given your mechanism I think it is necessary to discuss the differences to the mechanism described by Lehner et al 2013 as these authors suggest a mechanism which does not rely on the NAO. This is even more important as these authors use the same model though in different resolution.

We expanded the discussion explaining the differences and similarities of the mechanism described in Lehner et al. (2013) and our manuscript.

The new text is: " Another possible mechanism explaining cold events in the North Atlantic region is discussed by Lehner et al. (2013). In an ensemble of transient simulations from the Medieval Climate Anomaly to the Little Ice Age they used artificial sea ice growth as a sensitivity parameter. The sea ice is then transported to the sub-polar North Atlantic, melts, and reduces the deep-water formation and therefore the AMOC. Due to the reduced northward heat transport sea ice can expand and the northern North Atlantic and Nordic Seas cool further. The authors suggest that the sea ice transport is supported by increasing SLP above the Barents Sea but not by a NAO-anomaly. This mechanism hence differs from the mechanism presented here, where a NAO-anomaly triggers changes in winds and surface ocean circulation as well as freshwater transports."

Section 4.3: These oscillations are interesting but how model dependent are they? Is this behavior found also in other simulations with the same model? Is the reason mainly the coarse resolution and or the fact that the AMOC is rather weak and close to a threshold?

Yoshimori et al. (2010) also found an oscillatory mode in the low-resolution CCSM3 and described it in detail. In their study they detect a persistent cold mode with a weak AMOC and quasi-decadal oscillations. In addition to that, the AMOC in Stocker et al. (2007) also showed an oscillatory behavior. In our study the AMOC has a mean of ~12.3 Sv during the first cold event which could be an indicator for an AMOC threshold that is followed by a climate transition in the North Atlantic region. On the other hand, a role for the model resolution cannot be fully ruled out as we are not aware of such oscillations in higher resolution version of CCSM3 (which doesn't mean they don't exist).

We assume that the oscillations can be a signal for a nearby threshold and therefore nearby tipping point, since they have been described by Yoshimori et al. (2010) and Scheffer et al. (2009) to be interpreted as a property of the new 'oscillatory' mode. We expanded the section 4.3 in our manuscript accordingly.

The new text in section 4.3 is: A detailed description of this cold oscillatory mode in a low-resolution version of the CCSM3 was presented by Yoshimori et al. (2010). They presented the quasi-decadal oscillations as a feature of a cold climate mode with a weak AMOC. Furthermore, in the experiments by Stocker et al. (2007) one can see a similar oscillatory behavior for the AMOC. Therefore, we suggest that the oscillatory behavior during the events indicates a nearby tipping point towards the same cold mode."

P8,l20: "who showed "
done

Section 5: Given the fact that the model simulates to much sea ice how does this affect your proposed mechanism? Or in other words, is this mechanism only possible because of the rather strong biases in sea ice distribution?

Sea ice affects the ocean-atmosphere interaction and is therefore amplifying the cold events. However, we see no reason why the ultimate triggers of the cold events (i.e. atmospheric variability linked to NAO, changes in the SPG and oceanic freshwater transports) should be critically affected by the sea ice biases. We revised the last paragraph of Section 5 to clarify this point.

Fig. 1: Please start the caption with "Areas of "
done

Fig.2: The reference period seems to be selected in a rather warm period and roughly 1000 years after the second extreme event – why? I suggest to use a longer ref period just between the two extreme events so from 4000-3200. Or just two ref periods before or after the extreme event to avoid any orbital forcing signal.

We agree that the chosen reference period is a rather warm period and therefore changed the reference period as suggested to be the time span from 4000-3201 BP. The Figs. 2, 3, 6, 7, and 9 and all corresponding values have been changed accordingly (in the abstract and section 3.1, 3.2, and 4.2).

Fig.3: It remains unclear how the anomalies are calculated. I guess it is the difference between mean over the period of an event and reference. In this case I would call it a difference4 and not an anomaly (which is normally a deviation to a long-term mean, rather than a difference between independent periods). Please change this throughout the text.
done
Yes, it is the difference between mean over the period of an event and the reference period. This has been clarified in section 4.2 and the figure titles of Figs., 3, 6, and 9.

Fig. 6 and 9: see Fig.3

done, caption has been changed.

References:

- Dickson, R.R., Meincke, J., Malmberg, S.-A., Lee, A., 1988: The "great salinity anomaly" in the Northern North Atlantic 1968-1982. Prog. Ocean., 20,2, 103-151, doi:10.1016/0079-6611(88)90049-3.

- Lehner, F., Raible, C. C. & Stocker, T. F., 2012: Testing the robustness of a precipitation proxy-based North Atlantic Oscillation reconstruction. Quat. Sci. Rev. 45, 85–94.

- Lehner, F., Born, A., Raible, C.C., Stocker, T.F., 2013: Amplified inception of European Little Ice Age by sea ice-ocean-atmosphere feedbacks. Journal of Climate 26, 7586-7602.

- Ortega, P.,  Lehner, F., Swingedouw, D., Masson-Delmotte, V., Raible, C.C., Casado, M., Yiou, P., 2015, A model-tested North-Atlantic Oscillation reconstruction for the last millennium. Nature, 523 71-75.

- Raible, C. C., Lehner, F., Gonza´lez-Rouco, J. F. & Fernández-Donado, L., 2014: Changing correlation structures of the Northern Hemisphere atmospheric circulation from 1000 to 2100 AD. Clim. Past 10, 537–550.

- Scheffer, M., Bascompte, J., Brock, W.A., Brovkin, V., Carpenter, S.R.,  Dakos, V., Held, H., van Nes, E.H., Rietkerk, M.,  Sugihara, G., 2009: Early-warning signals for critical transitions, Nature, 461(7260), 53_-, doi:10.1038/nature08227.

- Schmutz, C., Luterbacher, J., Gyalistras, D., Xoplaki, E. & Wanner, H., 2000: Can we trust proxy-based NAO index reconstructions? Geophys. Res. Lett.27,1135–1138.

- Stocker, T.F., Timmermann, A., Renold, M., Timm, O., 2007: Effects of salt compensation on the climate model response in simulations of large changes of the Atlantic meridional overturning circulation. J. Clim. 20(24):5912–5928.

[Figure]

Figure R1: Zoom into the unreported event. Time series of the annual mean a) sea-surface temperature (SST; in °C) in the NWA, b) sea-surface salinity (SSS; in PSU) in the NWA, c) sea ice concentration in % in the Iceland Basin (green) and in the North West Atlantic (NWA; blue), d) mixed-layer depth (MLD; in m) in the NWA, e) AMOC calculated as the maximum of the streamfunction in the Atlantic north of 30°N and below 500 m water depth.

[Figure]

Figure R2: Difference maps for the unreported event of mixed-layer depth (m; a-b), sea ice concentration (%; c-d), sea surface temperature (°C; e-f) and sea surface salinity (PSU; g-h).

---

## Author Comment (AC2) · 30 Jun 2018

**Response to the interactive comments by E. Moreno-Chamarro:**

**Ref.: Ms. No. cp-2017-106**

**Title: Abrupt cold events in the North Atlantic in a transient Holocene simulation"**

We highly appreciate your time and effort giving constructive comments and suggestions, which will help us to greatly improve our manuscript. We have prepared a new version of the manuscript with the comment taken into account. Below the original comment is marked in black and our response in blue.

This is indeed an interesting paper. I don't aim at giving an exhaustive review here, but I would suggest that the authors compare and discuss their results with those in Moreno-Chamarro et al. [2015]. My impression is that many of the features and mechanisms of the cold events described here with the CCSM3 model are actually very similar to those of the decadal cold events in the simulations with the MPI-ESM model in that referred paper. For example, the conspicuous cooling and sea ice expansion around the Labrador Sea, the weakening of the subpolar gyre and the oceanic deep convection shutdown, or the length of the cold events itself. I would add, nonetheless, that the authors are here able to go a step farther and identify a potential trigger mechanism of such events. Such discussion would be a very valuable contribution to the Klus et al.'s manuscript.

We agree that the mechanism described in Moreno-Chamarro et al. (2015) features a lot of similarities to the mechanism presented in our study. The authors describe and analyze decadal cold events in the North Atlantic in climate simulations and climate reconstructions. Although being triggered by internal variability the external forcing can strengthen their behavior. Moreno-Chamarro et al. explain that the events start with a weakening of the sub-polar gyre, leading to a surface freshening and cooling followed by a shutdown of deep-convection. In our study the surface conditions in the sub-polar gyre also show a freshening and cooling and a weakening the deep-water formation in the northwest Atlantic. Furthermore, we also noticed a weakened sub-polar gyre. In addition we found anomalies in the freshwater transport through Denmark Strait and the Canadian Arctic Archipelago, potentially triggered by a prolonged positive phase of the NAO. This is a major difference to the study by Moreno-Chamarro et al. (2015) since they explicitly excluded an anomalous Arctic freshwater contribution as part of the trigger for the cold events. Still, both mechanisms show a plausible sequence and mechanisms for cold events in this sensitive region. We added the corresponding comparison to the discussion in section 4.2.

References:

- Moreno-Chamarro, E., Zanchettin, D., Lohmann, K., Jungclaus, J. H. (2015). Internally generated decadal cold events in the northern North Atlantic and their possible implications for the demise of the Norse settlements in Greenland. Geophysical Research Letters, 42(3), 908-915.